# Plasma Zonulin Levels as a Non-Invasive Biomarker of Intestinal Permeability in Women with Gestational Diabetes Mellitus

**DOI:** 10.3390/biom9010024

**Published:** 2019-01-11

**Authors:** Esra Demir, Hanise Ozkan, Kerem Doga Seckin, Berrak Sahtiyancı, Bulent Demir, Omur Tabak, Abdülbaki Kumbasar, Hafize Uzun

**Affiliations:** 1Health Sciences University, Kanuni Sultan Süleyman Training and Research Hospital, Internal Medicine Clinic, 34303 Istanbul, Turkey; esracokicli@hotmail.com (E.D.); haniseozkan@hotmail.com (H.O.); Berrak8@hotmail.com (B.S.); omurtabak@yahoo.com.tr (O.T.); abakikumbasar@yahoo.com (A.K.); 2Health Sciences University, Kanuni Sultan Süleyman Training and Research Hospital Gynecology and Obstetrics Clinic, 34303 Istanbul, Turkey; doga_seckin@hotmail.com; 3Health Sciences University, Bakirkoy Dr Sadi Konuk Training and Research Hospital Cardiology Clinic, 34147 Istanbul, Turkey; drbdmr06@hotmail.com; 4Department of Biochemistry, Cerrahpasa Medical Faculty, Istanbul University-Cerrahpasa, 34098 Istanbul, Turkey

**Keywords:** zonulin, gestational diabetes mellitus, pregnancy, intestinal permeability

## Abstract

Objective: We aimed to compare the levels of plasma zonulin, a non-invasive biomarker of increased intestinal permeability, between pregnant subjects, with and without gestational diabetes mellitus (GDM), at 24–28 gestational weeks. The eighty-five consecutive pregnant subjects that presented to our hospital’s obstetrics outpatient clinic and were diagnosed with GDM, for the first time by an oral glucose tolerance test (OGTT), formed the GDM group; 90 consecutive subjects that were not diagnosed with GDM by OGTT, formed the control group. The diagnosis of GDM was made by an OGTT performed between the 24th and 28th weeks of gestation, and in compliance with the American Diabetes Association (ADA) criteria. Plasma zonulin levels were measured by the enzyme-linked immunosorbent assay (ELISA) methods. The Plasma zonulin level was significantly higher in the GDM group than the control group (*p* < 0.001). A correlation analysis showed that plasma zonulin level was positively correlated to body mass index (BMI), creatinine, fasting plasma glucose, baseline, first hour, and two hours glucose levels and the OGTT, hemoglobin A1C (HbA1_C_), homeostatic model assessment for insulin resistance (HOMA-IR), and alanine aminotransferase (ALT) levels. Our findings suggest that zonulin may be a non-invasive biomarker involved in the pathogenesis of GDM. Further large-scale studies are needed on this subject.

## 1. Introduction

Gestational diabetes mellitus (GDM) is an important source of morbidity and mortality for both mother and infant [1]. Higher blood glucose levels, adversely affect the fetus’s intrauterine development. The GDM also increases the risk of diabetes mellitus (DM) development in later stages of life [1]. Hence, with respect to public health, it is of utmost importance to better elucidate the pathogenesis and risk factors of GDM.

In recent years it has been shown that gut microbiota plays a role in establishing body hemostasis and affects many systems, most notably the immune system. Prior studies have indicated that impaired gut microbiota plays a role in the pathogenesis of various disorders, most notably autoimmune disorders like inflammatory bowel disease, as well as DM, obesity, coronary artery disease (CAD), and colorectal cancer [2]. Furthermore, a disrupted intestinal mucosa, leading to increased intestinal permeability, is also thought to play a role in disease pathogenesis [3]. Zonulin is the basic protein that modulates tight junctions to regulate intercellular passage [4]. Human zonulin (47-kDa protein), also known as prehaptoglobin-2, binds to the epidermal growth factor receptor via protease-activated receptor 2 activation [5]. Zonulin is secreted mainly from the liver, but also from enterocytes, adipose tissue, brain, heart, immune cells, lungs, kidney, and skin [6,7]. Gliadin and bacteria induce zonulin secretion, which increases intestinal permeability, introducing foreign antigens to the immune system and triggering inflammation [8].

Former studies have shown that zonulin may play a role in the pathogenesis of various disorders, most notably celiac disease, as well as DM and CAD [5]. However, there is a paucity of information in the literature as to the relationship between GDM and zonulin. Herein, we aimed to compare the levels of plasma zonulin, an important marker of increased intestinal permeability, between pregnant subjects with and without GDM, at 24–28 gestational weeks.

## 2. Materials and Method

Our study was designed as a prospective observational study. Eighty-five consecutive pregnant subjects that presented to our hospital’s obstetrics outpatient clinic and were diagnosed with GDM, for the first time, by an oral glucose tolerance test (OGTT), formed the GDM group; 90 consecutive subjects that were not diagnosed with GDM by OGTT, formed the control group. Primary objective of study was to compare the two groups in terms of the plasma zonulin level. The secondary objective was to determine other parameters potentially affecting the plasma zonulin level.

All pregnant subjects that participated in the study gave informed consent, and the study was approved by our hospital’s ethics committee (Approval No: KAEK/2018.5.26). This study was conducted in accordance with the Declaration of Helsinki.

Exclusion criteria included subject’s refusal to participate, a pre-gestational diagnosis of DM, a history of GDM, autoimmune disorders, hepatic or renal dysfunction (creatinine level > 1.5 mg/dL), preeclampsia, active infection, and thyroid dysfunction.

The diagnosis of GDM was made by an OGTT performed between the 24th and the 28th weeks of gestation, and in compliance with the American Diabetes Association (ADA) guidelines [9]. Accordingly, a single-stage 75 g OGTT was used for subjects who were 24–28 weeks pregnant and free of a history of GDM. Following a fasting period of at least 8 h, fasting plasma glucose level was measured, which was followed by the administration of 75 g glucose and measuring the first and second hour glucose levels. GDM was diagnosed on the basis of the following criteria:(i)Fasting blood glucose: ≥92 mg/dL, or(ii)First hour plasma glucose: ≥180 mg/dL, or(iii)Second hour plasma glucose: ≥153 mg/dL.

Weight and body mass index (BMI) were recorded just before the OGTT. So, it was definitely pregnancy weight.

All subjects had at least 12 h of fasting before blood sampling for biochemical analysis at 24–28 gestational weeks. Blood samples were processed in a centrifuge at 3000 rpm (revolutions per minute). Plasma was collected and stored at −80 °C, until analysis. Plasma glucose, lipid profile aspartate aminotransferase (AST), alanine aminotransferase (ALT), gamma gluthamyl transferase (GGT), and uric acid levels were measured spectrophotometrically using the Abbot Aeroset 2.0 (Abbot Diagnostic, Abbott Park, IL USA). HbA_1c_ (%) was measured by the Variant 2 Turbo (Biorad, Hercules, CA, USA) which uses the glycation specific binding of boronated-affinity, to detect all the glycated Hb species present. Insulin levels were measured by the Cobas 8000 C702 (Roche Diagnostics, Indianapolis, IN, USA) chemistry analyzer; complete blood count was measured by the Sysmex XN 9000 (Symex Europe GmbH, Norderstedt, Germany) hematology analyzer. Electrolytes were measured by the Cobas 8000 C702 (Roche Diagnostics) chemistry analyzer. Homeostatic model assessment for insulin resistance (HOMA-IR) was calculated by the following formula: HOMA-IR = fasting glucose (mg/dL) × insulin (μIU/mL)/405.

### Measurement of Plasma Zonulin Levels 

Plasma zonulin levels were measured using the sandwich-enzyme-linked immunosorbent assay method, with the human enzyme-linked immunosorbent assay (ELISA) kit (Elabscience, Catalog Number: E-EL-H5560, Wuhan, Hubei Province, China). The coefficients of intra and inter-assay variation were 4.5% (*n* = 15) and 5.6% (*n* = 15), respectively.

## 3. Statistical Analysis

Study data were analyzed with Windows (SPSS Inc, Chicago, IL, USA) SPSS software package v. 22.0. The descriptive statistics included mean, standard deviation, median, minimum, maximum, frequency, and rate (%). The distribution of the variables was tested by the Kolmogorov-Smirnov test. Independent quantitative variables were analyzed using the *t* test and the Mann Whitney-U test. Independent qualitative variables were analyzed using the Chi-square test; when the criteria for the Chi-square test were not met; independent qualitative variables were compared using the Fisher test. Correlation analyses were carried out using the Spearman’s correlation analysis. Receiver operating characteristics (ROC) curves were used to calculate the cut-off points for the variables, to differentiate between GDM, with maximum sensitivity and specificity. A *p* value of less than 0.05 was considered statistically significant for all analyses.

## 4. Results

The demographics, clinical features, and laboratory parameters of the pregnant subjects are summarized in Table 1 and Table 2. There was no significant difference between the GDM and the control groups, with respect to age, height, hypertension, hyperlipidemia, smoking, and family history (for all comparisons *p* > 0.05) (Table 1).

Fasting plasma glucose, first hour, and two hour glucose levels in the OGTT, and HbA1C, HOMA-IR, and ALT levels were significantly higher in the GDM group, compared to the control group (for all comparisons *p* < 0.05) (Table 2). Moreover, plasma zonulin level was significantly higher in the GDM group than the control group (*p* < 0.001) (Table 2) (Figure 1).

A correlation analysis showed that the plasma zonulin level was positively correlated to BMI, creatinine, glucose, fasting, first hour, and two hour glucose levels in the OGTT, and HbA1C, HOMA-IR, and the ALT levels (for all correlations *p* < 0.05) (Table 3). There was a negative correlation between the zonulin and the HDL level (*p* < 0.05) (Table 3).

A ROC analysis was performed to determine the predictive power of the plasma zonulin level for the GDM and showed that the plasma zonulin level of 20 ng/mL or above had a sensitivity of 98.8%, a specificity of 100%, a positive predictive value (PPV) of 98.8%, and a negative predictive value (NPV) of 100% (Figure 2).

## 5. Discussion

Zonulin is a proposed novel circulating marker for intestinal permeability, and its increased concentrations reflect an increased intestinal permeability women with GDM. The main finding of this study was that plasma zonulin levels tended to be elevated during pregnancy with higher blood glucose, in a positive correlation with BMI, creatinine, fasting plasma glucose, first hour, and two hour glucose levels in the OGTT, as well as HbA1C, HOMA-IR, and ALT. 

Mokkala et al. [10] demonstrated for the first time, an association between an increased early-pregnancy serum zonulin concentration and GDM, suggesting zonulin as a possible predictor for GDM. Similar to the results of Mokkala et al. [10], in our results, plasma zonulin levels were found to be increased in GDM. The group of women with GDM was 24–28 weeks pregnant, in our study. Plasma zonulin level of 20 ng/mL or above had a sensitivity of 98.8%, a specificity of 100%, a positive predictive value (PPV) of 98.8%, and a negative predictive value (NPV) of 100%. The patients of Mokkala et al. were 12.8 ± 2.5 weeks pregnant. The early-pregnancy serum zonulin concentration was higher in women who developed GDM at mid-pregnancy. Using the ROC curve analyses and the Youden index, the optimal cut-off value for serum zonulin were ≥43.3 ng/mL, in predicting mid-pregnancy GDM, with a sensitivity of 88% (95% CI (confidence index): 71–100%) and a specificity of 47% (95% CI: 33–58%). This concentration had a PPV of 29% (95% CI: 16–41%) and a NPV of 94% (95% CI: 85–100%). These differences in sensitivity and specificity may be due to the differences in the gestational week. The clinical significance of the results of both studies may be displayed as a contribution to increased risk for pregnancy complications, including gestational diabetes. 

This is the first time that plasma zonulin level was positively correlated to BMI, fasting plasma glucose, HbA1C, and HOMA-IR in GDM. Hyperglycemia causes tissue damage. Increased intestinal permeability is associated with increased inflammation and insulin resistance (IR), and therefore, influence GDM onset. Hyperglycemia causes tissue damage and fosters the development of endothelial dysfunction. The latter strictly correlates with insulin resistance and inflammation [11]. The correlation between the zonulin and glucose levels may indicate that increased glucose and insulin concentrations may induce or regulate the zonulin secretion from blood vessels, during pregnancy. Zonulin concentration has also been shown to correlate with glucose levels, dyslipidemia, inflammation, and insulin resistance, in type 2 diabetes (T2DM) and obesity [12,13]. Circulating zonulin has a potential role in the pathophysiology of T2DM and obesity. Thus, increased zonulin does not only reflect intestinal permeability, but might also reflect a reaction, secondary to inflammation, in GDM.

Circulating zonulin is strongly correlated with lactulose—mannitol urine ratio, a widely used clinical indicator of guts permeability [14]—and is considered to be a non-invasive biomarker for gut permeability [6,15,16]. Circulating zonulin originates from several different tissues [6,7]. However, the degree of increased intestinal permeability, during pregnancy, has to be determined, in order to identify the etiology behind and the source of the elevated zonulin levels. Ohlsson et al. [17] demonstrated the absence of any correlation between serum and feces levels of zonulin. Thus, feces zonulin may be more associated with intestinal permeability, since secretion of zonulin from the intestinal barrier my leak into the lumen. Increased zonulin levels are considered to be a marker of an impaired intestinal barrier. The potential mechanism through which zonulin could contribute to the onset of gestational diabetes might originate in the intestine.

Diet has an important role in regulating intestinal permeability and, in turn, the risk for metabolic disorders. The present study did not adjust for dietary factors, which is a limitation. Plasma zonulin level was positively correlated to BMI, in our study. However, the results in pregnancy were controversial. Houttu et al. [18] showed that serum zonulin levels, as a marker of intestinal permeability, were not statistically significantly different between overweight and obese pregnant women. Mokkala et al. [19] showed that intestinal permeability, evaluated by serum zonulin concentration, was related to metabolic endotoxemia, markers of inflammation, and glucose and lipid metabolism, in pregnant overweight women. These results suggest an important role of zonulin for intestinal permeability, in inducing adverse metabolic reactions, and thus, a potential effect on the health of both the mother and the child. By reinforcing the intestinal barrier, it may be possible to manipulate maternal metabolism, during pregnancy, with subsequent health benefits. Higher zonulin levels are associated with higher waist circumference, diastolic blood pressure, fasting glucose, and increased risk of metabolic diseases [19]. The richness and composition of the gut microbiota and the intake of n–3 PUFAs, fiber, and a range of vitamins and minerals in overweight pregnant women are associated with the serum zonulin concentration. Modifying the composition of the gut microbiota and diet could beneficially influence intestinal permeability and, thus, may affect maternal and child health [20]. Pregnancy is associated with major shifts in intestinal permeability with increased zonulin levels that may play an important role in the observed increases in gestational inflammation, thereby potentially contributing to the development of GDM. 

Our study also had a few limitations. First, is the small patient number; second, is that additional markers of gut permeability dysfunction, other than zonulin, were not measured. The third limitation is the lack of data in term of patients’ diet and nutritional status. Probably, plasma zonulin levels could be affected by a patient’s dietary content. Our final limitation is our study’s observational design. Therefore, further large-scale randomized studies are needed to determine the exact role of zonulin in the pathogenesis of GDM.

In conclusion, plasma zonulin levels were higher in women with GDM, and it also correlated with glucose, HbA1_C_, and HOMA-IR. Increased plasma zonulin has a potential role in the pathophysiology of GDM. Increased zonulin does not only reflect intestinal permeability, but may also reflect a reaction, secondary to inflammation and IR, in GDM. Plasma zonulin may be used as a non-invasive biomarker for both risk stratification and prediction of therapeutic outcomes in women with GDM. Our findings should be confirmed by additional research, to study the role of zonulin, by influencing the energy balance in all endothelial and epithelial surfaces, e.g., brain, intestinal epithelium, and lung tissue in women with GDM.

## Figures and Tables

**Figure 1 biomolecules-09-00024-f001:**
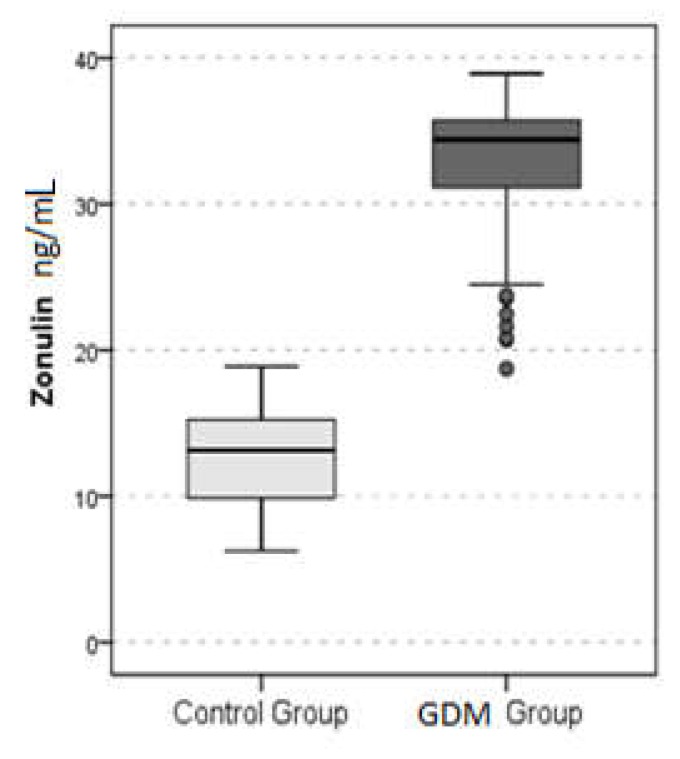
Plasma zonulin levels of the gestational diabetes mellitus (GDM) and control groups.

**Figure 2 biomolecules-09-00024-f002:**
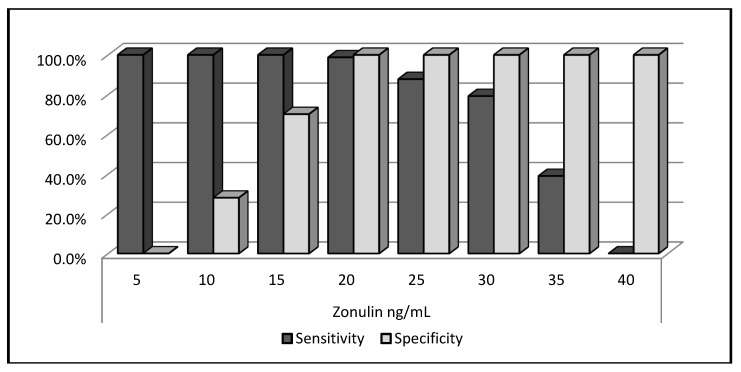
Receiver operator characteristic (ROC) curve analysis for zonulin.

**Table 1 biomolecules-09-00024-t001:** Demographic characteristics of the gestational diabetes mellitus (GDM) and control groups.

	Control Group			GDM Group			*p*
	Ort. ± s.d./*n*-%			Ort. ± s.d./*n*-%			
Age, years	30.2	±	4.9	31.3	±	6.4	0.169
Height, cm	160.8	±	5.7	161.8	±	5.5	0.206
Weight, kg	71.8	±	12.1	76.9	±	12.3	*0.004*
BMI; kg/m^2^	27.7	±	4.2	29.4	±	4.7	*0.027*
Pregnancy Number, n	2.3	±	1.3	2.7	±	1.4	0.077
History of GDM, n	6		5.1%	9		11.0%	0.120
HT, n	2		1.7%	2		2.4%	1.000
HL, n	0		0.0%	0		0.0%	1.000
Smoking, n	17		14.4%	13		15.9%	0.778
Family History, n	52		44.1%	29		35.4%	0.218

BMI—body mass index; GDM—gestational diabetes mellitus; HT—hypertension; HL—hyperlipidemia; s.d.—standard deviation; *n*—number; cm—centimeter; kg—kilogram; m^2^—square meter.

**Table 2 biomolecules-09-00024-t002:** The laboratory values of the GDM and control groups.

	Control Group			GDM Group				*p*
	Ort. ± s.d./*n*-%			Ort. ± s.d./*n*-%				
Glucose, mg/dL	72.5	±	11.0	84.9	±	14.6	84.0	0.000
Urea, mg/dL	14.3	±	4.0	14.6	±	4.0	14.0	0.356
Creatinine, mg/dL	0.4	±	0.1	0.4	±	0.1	0.4	0.148
AST, U/L	17.2	±	5.1	18.0	±	6.0	17.5	0.323
ALT, U/L	12.5	±	5.9	14.5	±	6.2	14.0	0.003
LDL, mg/dL	110.8	±	36.1	114.8	±	50.2	111.5	0.848
HDL, mg/dL	68.5	±	15.6	66.4	±	16.7	64.0	0.268
TG, mg/dL	195.5	±	77.9	208.7	±	69.8	200.5	0.083
T.Cholesterol, mg/dL	215.9	±	47.8	222.8	±	54.7	217.0	0.825
Uric acid, mg/dL	3.2	±	0.6	3.3	±	0.8	3.3	0.591
GGT, U/L	8.3	±	4.8	9.1	±	5.5	8.0	0.250
Hb, g/dL	11.2	±	1.3	11.2	±	1.0	11.2	0.868
WBC, (x10^3^)	9.9	±	2.3	10.2	±	2.1	10.1	0.268
Platelets (x10^9^)	241.2	±	69.8	239.0	±	61.1	233.5	0.805
MPV, fL	10.8	±	1.2	10.9	±	0.9	10.9	0.746
Sodium, mmol/L	137.4	±	1.4	137.2	±	1.7	138.0	0.716
Potassium, mmol/L	4.0	±	0.3	4.1	±	0.3	4.1	0.094
HbA1c, g/dL	4.8	±	0.4	5.0	±	0.4	5.0	0.004
HOMA-IR	2.2	±	1.1	2.5	±	1.0	2.4	0.004
Zonulin, ng/mL	12.8	±	3.3	32.6	±	4.8	34.4	0.000
OGTT 0, mg/dL	81.2	±	5.7	94.0	±	11.5	92.0	0.000
OGGT 1, mg/dL	128.2	±	24.3	173.5	±	33.9	175.0	0.000
OGGT 2, mg/dL	107.3	±	19.7	144.3	±	31.0	144.0	0.000

T.Chol—total cholesterol; LDL—low density lipoprotein; HDL—high density lipoprotein; TG—triglyceride; AST—aspratate aminotransferase; ALT—alanine aminotransferase; GGT—gamma-glutamyl transferase; Hb—hemoglobin; WBC—white blood counts; MPV—mean platelets volume; HbA1C—hemoglobin A1C; HOMA-IR—homeostatic model assessment-insulin resistance; OGTT–oral glucose tolerans test; s.d.—standard deviation; *n*—number; mg—milligram; dL—deciliter; U—unit; L—liter; g—gram; fL—femtoliter; ng—nanogram.

**Table 3 biomolecules-09-00024-t003:** Results of the Spearman correlation analysis.

		Age	BMI	Pregnancy Number	Glucose	Urea
**Zonulin**	r	0.041	0.199		0.361	−0.064
	*p*	0.633	0.019	0.463	0.000	0.455
		Creatinin	AST	ALT	LDL	HDL
**Zonulin**	r	0.247	0.020	0.197	0.039	−0.173
	*p*	0.003	0.815	0.020	0.646	0.041
		TG	T.Cholesterol	Uric Acid	GGT	Hg
**Zonulin**	r	0.140	0.028	0.150	0.096	−0.031
	*p*	0.100	0.746	0.078	0.259	0.719
		WBC	Platelets	MPV	Sodium	Potassium
**Zonulin**	r	0.102	0.018	0.040	0.014	0.126
	*p*	0.233	0.830	0.639	0.871	0.138
		HbA1c	HOMA-IR	OGTT 0	OGGT 1	OGGT 2
**Zonulin**	r	0.129	0.221	0.621	0.568	0.581
	*p*	0.131	0.009	0.000	0.000	0.000

BMI—body mass index; T.Chol—total cholesterol; LDL—low density lipoprotein; HDL—high density lipoprotein; TG—triglyceride; AST—aspratate aminotransferase; ALT—alanine aminotransferase; GGT—gamma-glutamyl transferase; Hg—hemoglobin; WBC—white blood counts; MPV—mean platelets volume; HbA1C—hemoglobin A1C; HOMA-IR—homeostatic model assessment-insulin resistance; OGTT—oral glucose tolerans test; *p*- *p* value; r- correlation coefficient.

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
