# Peer review of "Plasma Zonulin Levels as a Non-Invasive Biomarker of Intestinal Permeability in Women with Gestational Diabetes Mellitus"

_biomolecules, 2019, doi:10.3390/biom9010024_

Reviewer 1 Report

This is an interesting observational study comparing levels of plasma zonulin between women with and without GDM. Results indicate that levels of plasma zonulin were higher in women with GDM as compared to controls. In addition, women with GDM as compared to controls had higher 0, 1 and 2-hour glucose levels after the OGTT and higher HbA1C, HOMA-IR, and ALT levels.  While the topic of this study is of interest, the manuscript has several weaknesses.  

Major points:

1.       Abstract

Line 39-40. I don’t believe this conclusion is appropriate for this study “the increased plasma zonulin levels could reflect an enhanced intestinal permeability”. Authors have not measured intestinal permeability, only zonulin levels. The conclusion is that plasma zonulin levels (and other biochemical parameters) at 24-28 gestational weeks are significantly higher in women with GDM as compared to controls. It is true that increased intestinal permeability is associated with increased inflammation and insulin resistance, and therefore influence GDM onset. However, this is something to be discussed in the discussions section and should not be part of the conclusions.

2.       Introduction.

Line 68-69. The aim of the study needs to be more specific. For example, mention at some point that plasma zonulin levels were measured at 24-28 gestational weeks.

3.       Materials and Methods.

-          Primary and secondary objectives are not clearly stated in this section.

-          Could authors give more detail about how and when was demographic and anthropometric data recorded? Was it also recorded at 24-28 gestational weeks?

Moreover, when were weight and BMI recorded. Is it the pregestational or pregnancy weight and BMI? If weight was measured in pregnancy, at what time? This is important information that should be included in this section.

-          Diet seems an important confounder when it comes to gut microbiota and zonulin levels. In addition, diet is associated with GDM onset. Results in this study indicate that women with GDM had higher plasma zonulin levels. However, diet has not been considered and it is not possible to know whether this aspect influenced results. Did you register women’s dietary intake or nutritional habits at any point?

4.       Table 1.

It would be more appropriate to present demographic characteristics on a separate table. Specially since I am not sure whether these set of data (Table 1) have been collected at the same moment (24-28 gestational weeks), as pointed out previously.

In addition, it is very striking that mean BMI in both groups, even in the control group, is above 27 kg/m2. Are women of childbearing age mostly overweight in your geographical area? Or were only women of high risk included in this study? Moreover, I find it an important limitation that weight and BMI were significantly higher in women with GDM.  Zonulin can be also affected by BMI. So, is it possible that results could be confounded by the significantly higher BMI in the GDM group? (Kim JH et al. Clin Chim Acta. 2018;481:218-224. doi: 10.1016/j.cca.2018.03.005)

5.       Conclusions lines 222-223. Again, I do not believe that the conclusion of this study is that “increased plasma zonulin levels in patients with GDM may play a role in the development of GDM symptoms”. I think this is wrong. The conclusion should focus on the results found. Authors did not measure GDM symptoms. I recommend changing this sentence to one that mirrors the results of this study. In addition, the conclusions section needs to be more concise and summarize the key study outcomes and implications.

Minor points.

·       Line 59-60. I think the word “which” should be deleted. The sentence does not read well.

·       Line 83. The ADA stands by using two different criteria to diagnose GDM: one-step approach (using IADPSG criteria) and two-step approach (using either Carpenter-Coustan or NDDG criteria). The criteria used in this study are the IADPSG criteria. It sounds incorrect to define the IADPSG criteria as the ADA criteria. Maybe saying “ADA guidelines” is more correct. Or, to change ADA criteria to IADPSG criteria.

·       Line 121-122: “the patient ad control groups”. “ad” is missing and “n”. If you refer to the group of women with GDM as the “patient” group, this should be consistent throughout the manuscript. For instance, in the results sections the “patient” group is labelled as “GDM”. Please revise.  

·       Line 122: “height” is more appropriate than “body length”.

·       Line 131. What is the difference between “fasting plasma glucose” and “baseline glucose levels in the OGTT”? Aren’t they the same? I do not understand the difference.

·       Line 131. The OGTT can only be baseline, 1 hour and 2 hours post glucose load. I think “third hour glucose levels” is incorrect and should be changed to “two hour glucose levels”.

·       Line 132. “test” should be deleted.

·       Line 162. The word “that” should be deleted.

·       Line 180. “IR” has not been defined previously. Later, in line 182 authors refer to “insulin resistance”, in full. Please be consistent throughout the manuscript.

·       Table 1.

-          What does Ort.±s.s./n-% mean? Please specify in footnotes.

-          “Length” should be “height”.

-          How is DM diagnosis included in this table when this was one of the exclusion criteria?

-          “Hb” should be “Hg”?

Author Response

Dear Editor,

First, we would like thank the reviewers for the helpful comments, which led us to conduct appropriate experiments. The manuscript has subsequently been rewritten to address these concerns and comments of the reviewers.

We are grateful for your understanding and cooperation in this matter.

English

We have had our revised manuscript edited and proofread by a professional English-speaking editor to improve the readability and correct grammatical errors. We look forward to your reply. We believe that the language is now suitable for review.

Response to Reviewers:

Reviewer: 1

English language and style

(x) Extensive editing of English language and style required 
( ) Moderate English changes required 
( ) English language and style are fine/minor spell check required 
( ) I don't feel qualified to judge about the English language and style 

Yes

Can be improved

Must be improved

Not applicable

Does the introduction provide sufficient background and include   all relevant references?

( )

( )

(x)

( )

Is the research design appropriate?

( )

( )

(x)

( )

Are the methods adequately described?

( )

( )

(x)

( )

Are the results clearly presented?

( )

( )

(x)

( )

Are the conclusions supported by the results?

( )

( )

(x)

( )

Thank you so much for your valuable comments and suggestions.

Comments and Suggestions for Authors

This is an interesting observational study comparing levels of plasma zonulin between women with and without GDM. Results indicate that levels of plasma zonulin were higher in women with GDM as compared to controls. In addition, women with GDM as compared to controls had higher 0, 1 and 2-hour glucose levels after the OGTT and higher HbA1C, HOMA-IR, and ALT levels.  While the topic of this study is of interest, the manuscript has several weaknesses.  

Major points:

1.       Abstract

Line 39-40. I don’t believe this conclusion is appropriate for this study “the increased plasma zonulin levels could reflect an enhanced intestinal permeability”. Authors have not measured intestinal permeability, only zonulin levels. The conclusion is that plasma zonulin levels (and other biochemical parameters) at 24-28 gestational weeks are significantly higher in women with GDM as compared to controls. It is true that increased intestinal permeability is associated with increased inflammation and insulin resistance, and therefore influence GDM onset. However, this is something to be discussed in the discussions section and should not be part of the conclusions.

We performed the necessary changes in the abstract section (line: 46-48), (line: 185-186).

2.       Introduction.

Line 68-69. The aim of the study needs to be more specific. For example, mention at some point that plasma zonulin levels were measured at 24-28 gestational weeks.

We made the necessary changes in the introduction section (line: 87-88).

3.       Materials and Methods.

-          Primary and secondary objectives are not clearly stated in this section.

The primary objective is to compare the two groups in terms of plasma zonulin level.

The secondary objective is to determine other parameters which could affect zonulin level (line: 94-96).

-          Could authors give more detail about how and when was demographic and anthropometric data recorded? Was it also recorded at 24-28 gestational weeks?

Whole demographic and anthropometric data were recorded from pregnant women who were 24-28 weeks  (line: 115).

Moreover, when were weight and BMI recorded. Is it the pregestational or pregnancy weight and BMI? If weight was measured in pregnancy, at what time? This is important information that should be included in this section.

Weight and BMI were recorded just before the OGTT. So, it was definitely pregnancy weight (line: 114-115).

-          Diet seems an important confounder when it comes to gut microbiota and zonulin levels. In addition, diet is associated with GDM onset. Results in this study indicate that women with GDM had higher plasma zonulin levels. However, diet has not been considered and it is not possible to know whether this aspect influenced results. Did you register women’s dietary intake or nutritional habits at any point?

Thanks for this good question. Unfortunately, we did not have any data about patients’ diet and nutritional habits. This issue can be considerated as a limitation for our study. In the future; we plan to design a new study which will investigate the relationship between diet and zonulin. Thank you for your benefical contribution (line: 227-233).

4.       Table 1.

It would be more appropriate to present demographic characteristics on a separate table. Specially since I am not sure whether these set of data (Table 1) have been collected at the same moment (24-28 gestational weeks), as pointed out previously

In accordance with your suggestion, Table 1 was divided into two new tables. As we mentioned before, all data was collected just before the OGTT (line: 298-340).

In addition, it is very striking that mean BMI in both groups, even in the control group, is above 27 kg/m2. Are women of childbearing age mostly overweight in your geographical area? Or were only women of high risk included in this study? Moreover, I find it an important limitation that weight and BMI were significantly higher in women with GDM.  Zonulin can be also affected by BMI. So, is it possible that results could be confounded by the significantly higher BMI in the GDM group? (Kim JH et al. Clin Chim Acta. 2018;481:218-224. doi: 10.1016/j.cca.2018.03.005)

This is absolutely a good question, you are right about the GDM group having a higher BMI than the control group. We included the patients consecutively, and in our humble opinion, it is just a co-incidence. We also added BMI as an issue of limitation in the limitations section of our manuscript (line: 227-233).

5.       Conclusions lines 222-223. Again, I do not believe that the conclusion of this study is that “increased plasma zonulin levels in patients with GDM may play a role in the development of GDM symptoms”. I think this is wrong. The conclusion should focus on the results found. Authors did not measure GDM symptoms. I recommend changing this sentence to one that mirrors the results of this study. In addition, the conclusions section needs to be more concise and summarize the key study outcomes and implications.

We made the necessary changes in the discussion section (line: 234-238).

Minor points.

·       Line 59-60. I think the word “which” should be deleted. The sentence does not read well.

We made the necessary changes in the introduction section (line: 77).

·       Line 83. The ADA stands by using two different criteria to diagnose GDM: one-step approach (using IADPSG criteria) and two-step approach (using either Carpenter-Coustan or NDDG criteria). The criteria used in this study are the IADPSG criteria. It sounds incorrect to define the IADPSG criteria as the ADA criteria. Maybe saying “ADA guidelines” is more correct. Or, to change ADA criteria to IADPSG criteria.

‘’ADA criteria’’ has been changed as “ADA guidelines” (line: 105).

·       Line 121-122: “the patient ad control groups”. “ad” is missing and “n”. If you refer to the group of women with GDM as the “patient” group, this should be consistent throughout the manuscript. For instance, in the results sections the “patient” group is labelled as “GDM”. Please revise.  

We made the necessary changes (line: 147).

·       Line 122: “height” is more appropriate than “body length”.

We made the necessary changes in the results section (line: 147).

·       Line 131. What is the difference between “fasting plasma glucose” and “baseline glucose levels in the OGTT”? Aren’t they the same? I do not understand the difference.

They are the same.

·       Line 131. The OGTT can only be baseline, 1 hour and 2 hours post glucose load. I think “third hour glucose levels” is incorrect and should be changed to “two hour glucose levels”.

‘’ Third hour glucose levels’’ has been changed as “two hour glucose levels” (line: 45, 149, 154, 166).

·       Line 132. “test” should be deleted.

All ‘’ OGTT test’’ has been changed as “OGTT”. ·       

Line 162. The word “that” should be deleted.

We made the necessary changes.

·       Line 180. “IR” has not been defined previously. Later, in line 182 authors refer to “insulin resistance”, in full. Please be consistent throughout the manuscript.

We made the necessary changes. (line: 125-126).

·       Table 1.

-          What does Ort.±s.s./n-% mean? Please specify in footnotes.

-          “Length” should be “height”.

-          How is DM diagnosis included in this table when this was one of the exclusion criteria?

-          “Hb” should be “Hg”?

We made the necessary changes.

Reviewer 2 Report

The manuscript is not perfect in a format. All main text figures starting from low resolutions Figure 1 and Figure 2. I am unable to properly evaluate this manuscript due to the bothering information in Table 1 and Table 2. Please correct this. Let me reveiw the improved manuscript as revised mu comes.

Author Response

Dear Editor,

First, we would like thank the reviewers for the helpful comments, which led us to conduct appropriate experiments. The manuscript has subsequently been rewritten to address these concerns and comments of the reviewers.

We are grateful for your understanding and cooperation in this matter.

English

We have had our revised manuscript edited and proofread by a professional English-speaking editor to improve the readability and correct grammatical errors. We look forward to your reply. We believe that the language is now suitable for review.

Response to Reviewers:

Reviwer: 2

Open Review

English language and style

( ) Extensive editing of English language and style required 
( ) Moderate English changes required 
( ) English language and style are fine/minor spell check required 
(x) I don't feel qualified to judge about the English language and style 

Yes

Can be improved

Must be improved

Not applicable

Does the introduction provide sufficient background and include   all relevant references?

( )

( )

(x)

( )

Is the research design appropriate?

( )

( )

(x)

( )

Are the methods adequately described?

( )

( )

(x)

( )

Are the results clearly presented?

( )

( )

(x)

( )

Are the conclusions supported by the results?

( )

( )

(x)

( )

Comments and Suggestions for Authors

The manuscript is not perfect in a format. All main text figures starting from low resolutions Figure 1 and Figure 2. I am unable to properly evaluate this manuscript due to the bothering information in Table 1 and Table 2. Please correct this. Let me reveiw the improved manuscript as revised mu comes.

Thank you so much for your valuable comments and suggestions. In accordance with your suggestion, Table 1 was divided into two new tables, Also, the resolution of the figures has been increased.

Round  2

Reviewer 1 Report

Thank you for your response and for making the suggested amendments. I just have a few minor issues.

Minor points.

·       Line 131. What is the difference between “fasting plasma glucose” and “baseline glucose levels in the OGTT”? Aren’t they the same? I do not understand the difference. In the prior revision, I pointed out this. Authors answered that these two are the same. If they are the same, I think authors should pick one to avoid misunderstandings. In addition, if they are the same, I believe Table 1 needs to be revised. “Glucose mg/dL” and “0 OGTT” have different values. Given all this, I think there should only one item of fasting glucose.

·       Line 194. “was” should be “were”.

·       Conclusions lines 255-257. I still do not agree with the conclusion “In conclusion, increased plasma zonulin has a potential role in the pathophysiology of GDM. Increased zonulin does not only reflect intestinal permeability, but may also reflect a reaction secondary to inflammation and IR in GDM”. I think there should be some objective reference to the main results (primary and secondary objectives) found in the study.  I think the first sentence should mention that plasma zonulin levels were higher in women with GDM and that they also correlated with… After stating this, add the part highlighted in grey if you wish.

·       Table 1.

-          Ort.±s.s./n-%. Thank you for explaining this. S.d. is standard deviation, with a “d” not a “t” at the end.

-          How is DM diagnosis included in this table when this was one of the exclusion criteria? I am still confused about this.

Author Response

Open Review

English language and style

( ) Extensive editing of English language and style required 
(x) Moderate English changes required 
( ) English language and style are fine/minor spell check required 
( ) I don't feel qualified to judge about the English language and style 

Yes

Can be improved

Must be improved

Not applicable

Does the   introduction provide sufficient background and include all relevant references?

(x)

( )

( )

( )

Is the research   design appropriate?

(x)

( )

( )

( )

Are the methods   adequately described?

(x)

( )

( )

( )

Are the results   clearly presented?

( )

(x)

( )

( )

Are the conclusions   supported by the results?

( )

(x)

( )

( )

Comments and Suggestions for Authors

Thank you for your response and for making the suggested amendments. I just have a few minor issues.

Minor points.

·       Line 131. What is the difference between “fasting plasma glucose” and “baseline glucose levels in the OGTT”? Aren’t they the same? I do not understand the difference. In the prior revision, I pointed out this. Authors answered that these two are the same. If they are the same, I think authors should pick one to avoid misunderstandings. In addition, if they are the same, I believe Table 1 needs to be revised. “Glucose mg/dL” and “0 OGTT” have different values. Given all this, I think there should only one item of fasting glucose.

 Dear Reviewer;

Thank you for valuable comments.

As we mentioned before they are absolutely same. There is a little difference between ''Glucose'' and 0 OGTT, I have  tried  to explain it you below.  0 hour OGTT meaning was  to measure fasting plasma glucose just  at the begining of the  OGTT test. However; The meaning of ''Glucose'' was  the fasting plasma glucose which was measured different day than OGTT day. Additionally; their measuring methods are  probably different. Please consider the ‘’Glucose’’ as a just routine biochemical test. We obtanied fasting'' glucose'' value while we were measuring other routine laboratuary  parameters, such as LDL, AST etc.  I hope that I could  have explained differences between 0 OGTT and Glucose.

 ·  Line 194. “was” should be “were”.

 ·       Conclusions lines 255-257. I still do not agree with the conclusion “In conclusion, increased plasma zonulin has a potential role in the pathophysiology of GDM. Increased zonulin does not only reflect intestinal permeability, but may also reflect a reaction secondary to inflammation and IR in GDM”. I think there should be some objective reference to the main results (primary and secondary objectives) found in the study.  I think the first sentence should mention that plasma zonulin levels were higher in women with GDM and that they also correlated with… After stating this, add the part highlighted in grey if you wish.

 ·       Table 1.

-          Ort.±s.s./n-%. Thank you for explaining this. S.d. is standard deviation, with a “d” not a “t” at the end.

- thank you for suggestion. It has been corrected.

-          How is DM diagnosis included in this table when this was one of the exclusion criteria? I am still confused about this.

Dear reviewer;  you are absolutely  right your comment related to DM which is definetely our exclusion criteria.Thanks for valulable attention.  We had put the DM in Table 1 by mistake,  Therefore, we have deleted this incorrect information from Table 1

Reviewer 2 Report

The authors addressed all matters. This manuscript might be acceptable for the publication.

Author Response

(The authors gave the same response as above.)
